# Desertomycin G, a New Antibiotic with Activity against *Mycobacterium tuberculosis* and Human Breast Tumor Cell Lines Produced by *Streptomyces althioticus* MSM3, Isolated from the Cantabrian Sea Intertidal Macroalgae *Ulva* sp.

**DOI:** 10.3390/md17020114

**Published:** 2019-02-12

**Authors:** Alfredo F. Braña, Aida Sarmiento-Vizcaíno, Ignacio Pérez-Victoria, Jesús Martín, Luis Otero, Juan José Palacios-Gutiérrez, Jonathan Fernández, Yamina Mohamedi, Tania Fontanil, Marina Salmón, Santiago Cal, Fernando Reyes, Luis A. García, Gloria Blanco

**Affiliations:** 1Departamento de Biología Funcional, Área de Microbiología, e Instituto Universitario de Oncología del Principado de Asturias, Universidad de Oviedo, 33006 Oviedo, Spain; afb@uniovi.es (A.F.B.); UO209983@uniovi.es (A.S.-V.); marinasalmon92@gmail.com (M.S.); 2Fundación MEDINA, Centro de Excelencia en Investigación de Medicamentos Innovadores Andalucía, Avda. del Conocimiento 34, Parque Tecnológico de Ciencias de la Salud, E-18016 Granada, Spain; ignacio.perez-victoria@medinaandalucia.es (I.P.-V.); jesus.martin@medinaandalucia.es (J.M.); 3Servicio de Microbiología, Hospital de Cabueñes, 33203 Gijón, Spain; oteroluis@uniovi.es; 4Servicio de Microbiología, Hospital Universitario Central de Asturias, 33011 Oviedo, Spain; juanjose.palacios@sespa.es (J.J.P.-G.); jofersua@hotmail.com (J.F.); 5Departamento de Bioquímica y Biología Molecular, e Instituto Universitario de Oncología del Principado de Asturias, Universidad de Oviedo, 33006 Oviedo, Spain; yamomu@hotmail.com (Y.M.); taniuskina@gmail.com (T.F.); santical@uniovi.es (S.C.); 6Departamento de Ingeniería Química y Tecnología del Medio Ambiente, Área de Ingeniería Química, Universidad de Oviedo, 33006 Oviedo, Spain; luisag@uniovi.es

**Keywords:** desertomycin, macrolide, *Streptomyces*, seaweed associated actinobacteria, actinomycetes, intertidal seaweed

## Abstract

The isolation and structural elucidation of a structurally new desertomycin, designated as desertomycin G (**1**), with strong antibiotic activity against several clinically relevant antibiotic resistant pathogens are described herein. This new natural product was obtained from cultures of the marine actinomycete *Streptomyces althioticus* MSM3, isolated from samples of the intertidal seaweed *Ulva* sp. collected in the Cantabrian Sea (Northeast Atlantic Ocean). Particularly interesting is its strong antibiotic activity against *Mycobacterium tuberculosis* clinical isolates, resistant to antibiotics in clinical use. To the best of our knowledge, this is the first report on a member of the desertomycin family displaying such activity. Additionally, desertomycin G shows strong antibiotic activities against other relevant Gram-positive clinical pathogens such as *Corynebacterium urealyticum*, *Staphylococcus aureus*, *Streptococcus pneumoniae*, *Streptococcus pyogenes*, *Enterococcus faecium*, *Enterococcus faecalis*, and *Clostridium perfringens*. Desertomycin G also displays moderate antibiotic activity against relevant Gram-negative clinical pathogens such as *Bacteroides fragilis*, *Haemophilus influenzae* and *Neisseria meningitidis*. In addition, the compound affects viability of tumor cell lines, such as human breast adenocarcinoma (MCF-7) and colon carcinoma (DLD-1), but not normal mammary fibroblasts.

## 1. Introduction

Desertomycins are a family of macrolide natural products of pharmacological interest due to their diverse biological activities. Structurally, they belong to the marginolactones family; aminopolyol polyketides containing a macrolactone ring whose biosynthesis is the subject of active research [1]. After a report on the first member of the family, desertomycin A, with antibiotic activity against bacteria and fungi, [2], other structurally related compounds of the family such as the desertomycins B and D [3], E [4] and F [5] have also been identified. They are mainly produced by *Streptomyces* species isolated from soils, such as *Streptomyces macronensis* and *Streptomyces flavofungini* [6], but have also been found in cultures of *Streptoverticillium baldaccii* [7].

The marine environment has become a prime resource in the search for, and discovery of, novel natural products. Marine actinomycetes (Phylum *Actinobacteria*) have turned out to be important contributors [8]. Marine macroalgae (seaweeds) remain a relatively underexplored source in the search of *Streptomyces* producing bioactive compounds of pharmacological interest. Previous reports describe the isolation of actinomycetes of the *Streptomyces* genus from seaweeds on coastal ecosystems from temperate and cold waters of the North Atlantic Ocean, particularly from the Iberian Peninsula coasts [9] and from the Kiel Fjord in the Baltic Sea [10,11]. Previous research in the Cantabrian Sea (Biscay Bay, Northeast Atlantic) has revealed that bioactive *Streptomyces* species are sometimes associated with marine macro-algae [12,13,14]. Oceanic and atmospheric environments in this region [13,14,15] are emerging as novel sources for the discovery of new natural products with antibiotic properties and cytotoxic activities towards cancer cell lines [16,17,18,19,20].

We report herein the discovery of a new natural product, desertomycin G (**1**), obtained from *Streptomyces althioticus* MSM3, isolated from samples of the intertidal seaweed *Ulva* sp. (Phylum *Chlorophyta*) collected in the Cantabrian Sea. After growing this strain in R5A medium, extracts of the culture broth exhibited a strong antibiotic activity. A bioassay-guided identification of the active fractions after a semipreparative HPLC analysis of an extract pointed to a single peak as responsible for the activity. Following our LC-UV-MS and LC-HRMS chemical dereplication approaches for marine natural products [21], this peak turned out not to be included in our library and prompted us to perform isolation and identification. Its structure was elucidated by HRMS and 1D and 2D NMR analyses. Desertomycin G displays potent antibiotic activities against Gram-positive clinical pathogens and, to the best of our knowledge, is the first member of the family displaying strong antibiotic activity against *Mycobacterium tuberculosis*. Additionally, desertomycin G shows cytotoxic activity against two human tumor cell lines.

## 2. Results

### 2.1. Taxonomy and Phylogenetic Analysis of the Strain MSM3

The 16 S rDNA of the producing strain MSM3 was polymerase chain reaction (PCR) amplified and sequenced as previously reported [14]. Sequence analysis showed a 99.9% identity to Streptomyces althioticus (AY999808). Thus, this strain was designated as *Streptomyces althioticus* MSM3 (EMBL Sequence number LT627193). Another strain of this species producing an unidentified antituberculous compound, *Streptomyces althioticus* (KCTC 9752), was isolated from the Thar Desert soil, in Rajasthan [22]. The phylogenetic tree generated by a neighbor-joining method based on 16S rRNA gene sequence clearly revealed the evolutionary relationship of the strain MSM3 with a group of known Streptomyces species (Figure 1).

### 2.2. Structure Determination

A molecular formula of C_62_H_109_NO_21_ was assigned to desertomycin G, according to a protonated ion observed at *m*/*z* 1204.7609 in its ESI-TOF spectrum (calcd. for C_62_H_110_NO_21_^+^ 1204.7565, ∆ 3.6 ppm). Analysis of its ^1^H, ^13^C (Table 1) and HSQC spectra revealed the presence in the molecule of 10 olefinic protons, 19 oxygenated methines, one doubly oxygenated methine, probably belonging to the anomeric position of a sugar residue, one oxygenated methylene, 8 aliphatic methines, 9 aliphatic and one nitrogenated methylenes and 10 aliphatic methyl groups, suggesting a polyketidic nature for the compound. Correlations observed in the COSY spectrum allowed us to establish the sequences from C-3 to C-19 and from C-21 to C-46 and also confirmed the attachment of methyl groups C-48, C-49, C-50, C-51, C-53, C-54, C-55 and C-56 to C-6, C-8, C-14, C-18, C-24, C-30, C-32 and C-42, respectively (Figure 2). On the other hand, HMBC correlations between H_3_-47 to C-1, C-2 and C-3, and H_3_-52 to C-19, C-20 and C-21 allowed completion of the linear sequence from C-1 to C-46. An additional HMBC correlation between H-41 and C-1 and the deshielded oxygenated proton H-41 (5.11 ppm) was also indicative of the existence of a lactone ring between C-41 and C-1 (Figure 2). Finally, the only nitrogen atom present in the molecule was associated with the presence of a primary amine group at C-46, as evidenced by the corresponding ^1^H and ^13^C NMR chemical shifts at this terminal position (δ_H_ 2.93 ppm and δc 40.8 ppm). A bibliographic search established that the planar structure of this macrocyclic moiety of the molecule was very similar to that found in desertomycin A, the two major differences between both molecules being the presence of an additional double bond at ∆^4^ and an additional methyl group (C-53), located at C-24 in the structure of desertomycin G. The *E* configuration of the ∆^4^ double bond was proposed based on the large *J*_H4-H5_ coupling constant measured (15.0 Hz). The remaining signals not yet assigned in the NMR spectra corresponded to one doubly oxygenated methine group, four oxygenated methines and one oxygenated methylene (C-1′ to -C-6′). These signals were in agreement with the presence in the molecule of an hexopyranose, identified as α-d-mannopyranose based on similar chemical shifts to those measured in the structure of desertomycin A [23]. Additionally, HMBC correlations from H-1′ to C-22 and from H-22 to C-1′ (Figure 2) confirmed the linkage of this sugar moiety to carbon C-22.

### 2.3. Antimicrobial Activity of Desertomycin G

The antimicrobial activity of compound **1** was tested against a panel of human pathogens (Table 2). Some of them were isolated and identified in clinical microbiology laboratories from samples obtained from patients with clinical infections. The compound exhibited strong inhibitory activities against the pathogenic Gram-positive bacteria *Corynebacterium urealyticum, Staphylococcus aureus, Streptococcus pneumoniae, Streptococcus pyogenes, Enterococcus faecium, Enterococcus faecalis, Clostridium perfringens* and *Mycobacterium tuberculosis*; and moderate inhibition of the Gram-negative bacteria *Bacteroides fragilis*, *Haemophilus influenzae,* and *Neisseria meningitidis.*

### 2.4. Cytotoxic Activity of Desertomycin G

The cytotoxic activity of compound **1** was examined against the A549 human lung carcinoma, DLD-1 colon carcinoma and MCF-7 human breast adenocarcinoma cell lines, as well as healthy mammary fibroblasts. At day three, results indicated that DLD-1 and MCF-7 cell lines decreased their viability about 50% with respect to the control when employing 2,5 and 5 µM desertomycin G. A549 was more resistant and healthy mammary fibroblasts remained unaffected at these concentrations (Figure 3).

## 3. Discussion

Natural products of the desertomycin family were isolated in the last century from soil samples. A new member of the family, desertomycin G, was herein identified from cultures of the marine derived *Streptomyces althioticus* MSM3, associated with the intertidal seaweed *Ulva* sp. from the Cantabrian Sea. Its structure was established by analysis of its HRMS and 1D and 2D NMR spectra. Due to its potent antibiotic activities against clinically resistant pathogens and, given the medical needs for novel antibiotics against these microorganisms, desertomycin G deserves to be considered as a candidate for antibiotic chemotherapy, particularly against *Mycobacterium tuberculosis* resistant strains, but also against the rest of the indicated pathogens. Its cytotoxicity against tumor cell lines, but not against mammary fibroblasts, also highlights the use of desertomycin G as a potential antitumor agent. This is another example of the potential of marine *Streptomyces* species, particularly those isolated from the Cantabrian Sea, as new sources for the discovery of novel natural products with great potential as both antibiotic and antitumor agents.

## 4. Materials and Methods

### 4.1. General Experimental Procedures

Analytical and semipreparative HPLC analyses were conducted using a Waters Alliance (Waters Corporation, Milford, MA, USA) chromatographic system with a SunFire C18 column (10 µm, 10 × 250 mm, Waters). For UPLC analysis an Acquity UPLC equipment with a BEH C18 column (1.7 μm, 2.1 × 100 mm, Waters) was used. Optical rotations were measured using a Jasco P-2000 polarimeter (JASCO Corporation, Tokyo, Japan). UV spectra were obtained with an Agilent 1100 DAD (Agilent Technologies, Santa Clara, CA, USA). IR spectra were recorded on a JASCO FT/IR-4100 spectrometer (JASCO Corporation, Tokyo, Japan) equipped with a PIKE MIRacle^TM^ single reflection ATR accessory (PIKE Technologies Inc., Madison, WI, USA). NMR spectra were recorded on a Bruker Avance III spectrometer (500 and 125 MHz for ^1^H and ^13^C NMR, respectively) equipped with a 1.7 mm TCI MicroCryoProbe^TM^ (Bruker Biospin, Falländen, Switzerland), using the signal of the residual solvent as internal reference (δ_H_ 3.31 and δ_C_ 49.0 ppm for CD_3_OD). HRESIMS spectra were acquired using a Bruker maXis QTOF spectrometer (Bruker Daltonik GmbH, Bremen, Germany).

### 4.2. Microorganism and Fermentation Conditions

*Streptomyces althioticus* strain MSM3 was isolated from the macroalgae *Ulva* sp. collected in the Cantabrian Sea in Pedreña, Cantabria (coordinates 43°26′37″N, 3°46′5″W). Thirty Erlenmeyer flasks (250 mL), each containing 50 mL of R5A medium [12] were inoculated with spores of this strain and incubated in an orbital shaker at 28 °C and 250 rpm for 6 days.

### 4.3. Isolation and Purification of Desertomycin G

The cultures were then centrifuged and the pellets were discarded. The supernatants were filtered and applied to a solid-phase extraction cartridge (Sep-Pak Vac C18, 10 g, Waters). The retained material was eluted with a mixture of methanol and 0.05% TFA in water. A linear gradient from 0 to 100% methanol in 60 min, at 10 mL/min, was used. Fractions were collected every 5 min and analyzed by UPLC using chromatographic conditions previously described [12]. A peak corresponding to the desired compound was observed in fractions taken between 35 and 45 min. These fractions were pooled, partially dried in vacuo, and applied to a to a solid-phase extraction cartridge (Sep-Pak Vac C18, 2 g, Waters). The cartridge was washed with water; the retained compounds were eluted with methanol and dried in vacuo. The residue was subsequently re-dissolved in a small volume of methanol and DMSO (1:1). Purification was performed in two steps using a SunFire C18 column (10 µm, 10 × 250 mm, Waters). In the first step the extract was chromatographed with a mixture of acetonitrile and 0.1% TFA in water (32:68) in isocratic conditions and a flow rate of 7 mL/min. In the second step the mobile phase was a mixture of methanol and 0.1% TFA in water (67:33), at 6 mL/min. After every step, the solution containing the collected peak was partially evaporated under vacuum to reduce the concentration of the organic solvent and then applied to a solid-phase extraction cartridge (Sep-Pak C18, 500 mg, Waters). The cartridge was washed with water and the retained compound was eluted with methanol. The purified compound was finally re-dissolved in a mixture of tert-butanol and water (1:1) and lyophilized, resulting in 20.4 mg of pure product.

Desertomycin G **(1)**: white solid; [α]^20^_D_ + 15.6 (c 0.35, MeOH); IR (ATR) *ν*_max_ 3367, 2970, 2927, 1683, 1637, 1455, 1433, 1387, 1255, 1232, 1203, 1136, 1105, 1070, 1053, 971 cm^−1^; for ^1^H and ^13^C NMR data see Table 1; HRESIMS *m*/*z* 1204.7609 [M + H]^+^ (calcd. for C_62_H_110_NO_21_^+^, 1204.7565), 593.8776 [M − H_2_O + 2H]^2+^ (calcd. for C_62_H_109_NO_20_^2+^, 593.8774).

### 4.4. Phylogenetic Analysis (Taxonomy) of the Producer Microorganism

Strain *Streptomyces althioticus* MSM3 was subjected to phylogenetic analysis based on 16S rRNA sequences analysis. Phylogenetic analysis was performed using MEGA version 6.0 [24] after multiple alignment of data by CLUSTALO [25]. Distances (distance options according to the Kimura two-parameter model [26]) and clustering with the neighbor-joining [27] method were determined using bootstrap values based on 1000 replications [28].

### 4.5. Antimicrobial Activity of Compound against Clinic Pathogens

The antimicrobial activities of the compound were assessed and the minimum inhibitory concentrations (MIC) were determined against a panel of human pathogens, some of them multi-resistant to clinically-used antibiotics (Table 2). Some of them were isolated and identified in clinical microbiology laboratories from samples obtained in patients with clinical infections. Mueller-Hinton agar (Becton, Dickinson and Company, Le Pont de Cloix, France) was the culture medium in bioassays against *E. coli*, *S. aureus*, *E. faecalis*, *E. faecium*, *M. luteus*, *H. influenzae*, being supplemented according to the CLSI conditions for *S. pneumoniae*, *S. pyogenes* and *N. meningitidis.* Trypticasein soy agar (5% *w*/*v*) sheep blood DIFCO^TM^ (Becton, Dickinson and Company) was used for *C. urealyticum.* Brucella Broth (SIGMA-ALDRICH, St. Louis, MO, USA) supplemented with hemin (5 µg/mL), vitamin K1 (1 µg/mL) and lysed horse blood (5% *v*/*v*) was used for *B. fragilis* and *C. perfringens*.

For most Gram-positive and Gram-negative bacteria, the antimicrobial assays were performed according to CLSI performance standards [29]. For *M. tuberculosis* susceptibility testing was done in Middlebrook 7H10 agar medium (Becton, Dickinson and Company) supplemented with 10% OADC and 0.5% glycerol according to the agar proportion method for slowly growing mycobacteria [30].

### 4.6. Cytotoxic Activity of Desertomycin G

Cell proliferation was measured using the CellTiter 96 Non-radioactive Cell Proliferation Assay kit purchased from Promega Promega Biotech Iberica, Alcobendas, Spain). To this end, 10^3^ cells were seeded in 96-well plates and six replicates per condition and time point were assessed. Cell proliferation rates were determined for five consecutive days using an automated microtiter plate reader Power Wave WS (BioTek, Bad Friedrichshall, Germany)). The conditions studied were A549, DLD-1, MCF-7 cancer cells, and healthy mammary fibroblasts in the presence or absence of different concentrations of desertomycin G.

## Figures and Tables

**Figure 1 marinedrugs-17-00114-f001:**
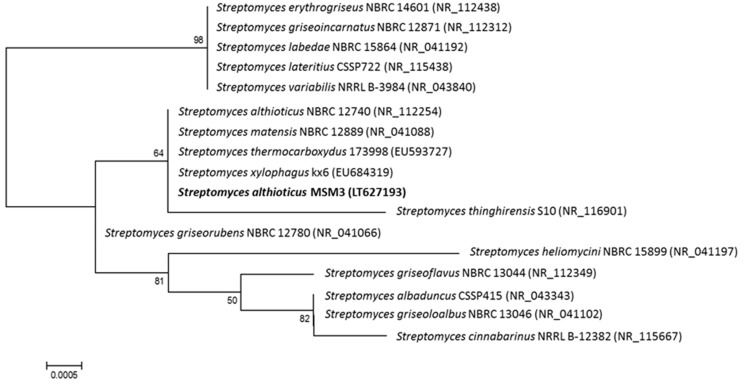
Neighbor-joining phylogenetic tree obtained by distance matrix analysis of 16S rDNA sequences, showing *Streptomyces althioticus MSM3* position and its most closely related phylogenetic neighbors. Numbers on branch nodes are bootstrap values (1000 re-samplings; only values >50% are given). Bar indicates 0.05% sequence divergence.

**Figure 2 marinedrugs-17-00114-f002:**
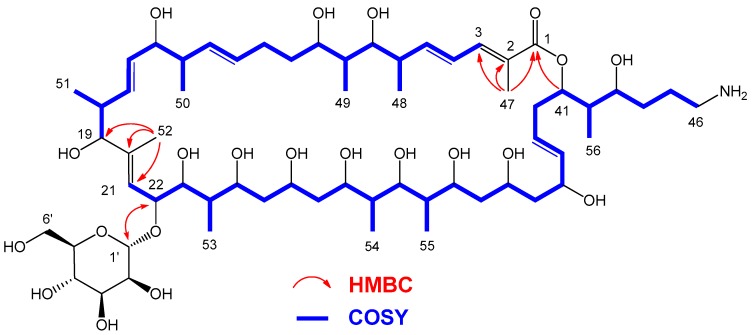
Key COSY and HMBC correlations determining the connectivity of **1**.

**Figure 3 marinedrugs-17-00114-f003:**
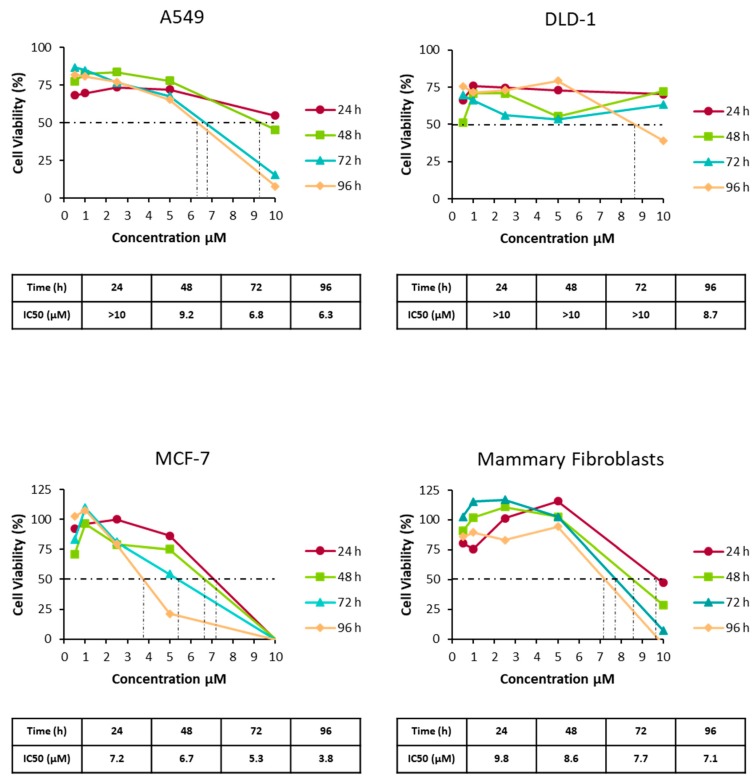
Cell proliferation of A549, DLD-1, MCF-7 cancer cells lines and healthy mammary fibroblasts in the presence or absence of different concentrations of desertomycin G. Cell proliferation rates were determined on five consecutive days using an automated microtitre plate reader. *** Significant differences with a *p* value less than 0.005.

**Table 1 marinedrugs-17-00114-t001:** ^1^H and ^13^C NMR assignments for desertomycin G (**1**) (CD_3_OD, 500 MHz).

	^1^H NMR	^13^C NMR		^1^H NMR	^13^C NMR
Position	δ in ppm (mult, *J* in Hz)	δ in ppm	Position	δ in ppm (mult, *J* in Hz)	δ in ppm
1		169.8	32	1.67 (m)	41.7
2		126.4	33	4.17 (br d, 9.7)	70.1
3	7.20 (br d, 11.1)	140.8	34	1.63 (m); 1.40 (m)	42.5
4	6.47 (dd, 14.4, 11.4)	126.2	35	4.00 (m)	66.4
5	6.23 (dd, 15.0, 7.5)	148.9	36	1.51 (m)	46.4
6	2.58 (m)	41.2	37	4.26 (m)	69.6
7	3.48 (dd, 8.7, 3.0)	78.6	38	5.57 (dd, 15.5, 4.8)	138.2
8	1.77 (m)	43.0	39	5.63 (m)	125.8
9	3.78 (m)	74.6	40	2.46 (m); 2.31 (m)	34.6
10	1.62 (m); 1.39 (m)	34.0	41	5.11 (m)	75.7
11	2.25 (m); 2.06 (m)	30.4	42	2.00 (m)	43.8
12	5. 48 (m)	131.5	43	3.53 (m)	72.5
13	5.48 (m)	134.1	44	1.65 (m); 1.40 (m)	30.3
14	2.19 (m)	44.0	45	1.83 (m); 1.67 (m)	25.6
15	3.87 (m)	76.8	46	2.93 (m)	40.8
16	5.49 (m)	132.1	47	1.94 (br s)	12.8
17	5.49 (m)	134.6	48	1.06 (d, 6.6)	12.8
18	2.35 (m)	41.2	49	0.85 (d, 6.5)	12.3
19	3.71 (m)	83.6	50	0.96 (m)	15.5
20		146.2	51	1.12 (d, 6.5)	17.7
21	5.40 (br d, 9.8)	122.9	52	1.76 (br s)	12.2
22	4.56 (dd, 9.9, 3.2)	72.7	53	0.85 (d, 6.5)	11.2
23	3.77 (m)	77.3	54	0.94 (m)	10.1
24	1.53 (m)	41.0	55	0.78 (d, 6.8)	11.5
25	4.29 (m)	68.9	56	0.95 (m)	10.6
26	1.71 (m); 1.40 (m)	42.8	1′	4.86 (m)	97.7
27	4.03 (m)	69.4	2′	3.77 (m)	72.3
28	1.72 (m)	43.4	3′	3.75 (m)	72.7
29	3.82 (m)	75.1	4′	3.63 (dd, 9.2)	68.8
30	1.63 (m)	40.9	5′	3.56 (m)	74.9
31	3.99 (m)	73.5	6′	3.85 (m); 3.73 (m)	62.9

**Table 2 marinedrugs-17-00114-t002:** Description of clinic bacterial pathogens and MIC values for compound **1**.

Clinical Pathogen	Isolate	Hospital	Year	Antibiotic Resistances	MIC (µg/mL)
**Gram-positives**					
*M. tuberculosis* H37Rv	ATCC 27294			-	16
*M. tuberculosis* MDR-1	14595	SNRL-Spain	2013	Multiresistance ^a^	16
*M. tuberculosis* MDR-2	14615	SNRL-Spain	2013	Multiresistance ^b^	16
*C. perfringens*	103281 *	HUCA	2013	-	16
*C. urealyticum*	1492 *	Cabueñes	2014	Multiresistance ^c^	<0.25
*E. faecalis*	10544	Cabueñes	2015	Ery, clin, tet	8
*E. faecalis*	ATCC 51299	-		-	8
*E. faecalis*	ATCC 29212	-		-	8
*E. faecium*	10701	Cabueñes	2015	Amp, quin, ery	4
*S. pneumoniae*	64412 *	HUCA	2013	Ery	
*S. pyogenes*	81293 *	HUCA	2013	-	
*S. aureus*	11497	Cabueñes	2015	Methicillin sensitive	4
*S. aureus*	ATCC 43300	-		-	4
S. aureus	ATCC 25923	-		-	4
**Gram-negatives**		-			
*B. fragilis*	61592 *	HUCA	2013	Amo, tet	32
*B. fragilis*	ATCC 25285	-		-	32
*H. influenzae*	10996	Cabueñes	2015	Amp, cot, quin	>64
*H.influenzae*	ATCC 49247	-		-	64
*N. meningitidis*	71327	HUCA	2013	Clin	64

* [14]; Amk: amikacin; amo: amoxicillin; amp: ampicillin; cap: capreomycin; cip: ciprofloxacin; clav: clavulanic acid; clin: clindamycin; cot: cotrimoxazole; emb: ethambutol; ery: erythromycin; fos: fosfomycin; inh: isoniazid; kan: kanamycin; nitro: nitrofurantoin; quin: quinolones; rif: rifampicin; str: streptomycin; tet: tetracycline. ^a^ Inh, rif, emb; ^b^ Inh, rif, emb, str, amk, kan, cap; ^c^ Amp, amo/clav, ery, cot, cip, fos, nitro.

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
