# Peer review of "Desertomycin G, a New Antibiotic with Activity against Mycobacterium tuberculosis and Human Breast Tumor Cell Lines Produced by Streptomyces althioticus MSM3, Isolated from the Cantabrian Sea Intertidal Macroalgae Ulva sp."

_marinedrugs, 2019, doi:10.3390/md17020114_

Reviewer 1 Report

This paper describes the isolation and structural determination of desertomycin G from cultures of the marine actinomycete Streptomyces althioticus MSM3, isolated from the intertidal seaweed Ulva sp. Collected in the Cantabrian Sea. The structural determination of this compound is appropriately carried out by spectroscopic techniques and comparing spectral data of this compound and that of known desertomycin A. Of particular note is that desertomycin G shows strong antibiotic activity against Mycobacterium tuberculosis clinical isolates, resistant to antibiotics in clinical use. In addition, desertomycin G shows antibiotic activities against Gram-positive and Gram-negative clinical pathogens. This compound also affects the survival rate of tumor cell lines such as human breast adenocarcinoma and colon carcinoma, but does not affect normal breast fibroblasts. As mentioned above, desertomycin G has very interesting biological activities.

This paper is interesting content and there is no mistake. I will recommend that this manuscript will be acceptable without revision.

Author Response

We appreciate the positive comments of this reviewer and the time dedicated to revise our manuscript.

Reviewer 2 Report

With the current study (marinedrugs-437154), the authors report the isolation and structure elucidation of a new desertomycin as well as its antibacterial and cytotoxic effects towards a series of cancer cell lines. The content is unequivocally aligned with the scope of the journal, i.e. the structural novelty of the compound and the preliminary results on its potential therapeutic utility. The acceptance of the manuscript is advised, but minor changes and suggestions should be considered by the authors.

While generally well-written, very minor modifications should be considered as in:

Line 28: Consider “…isolated from samples of the intertidal seaweed Ulva sp.”

Line 34: Enterococcus pyogenes should be deleted.

Line 59-61: According to the authors it appears that bioactives-producing Streptomyces spp. are solely and restrictively obtained from marine seaweeds, which is scientifically incoherent, as they are frequently isolated from other sources such as terrestrial ones. As such, I would recommend modifying to: ….bioactive Streptomyces species are frequently associated to marine macro-algae”. 

Lines 62-63: The term “cytotoxic” solely reflects cell toxicity and not directly any potential therapeutic utility if not towards cancer cells. As such, I would recommend modifying to: “…of new natural products with antibiotic properties and cytotoxic activities towards cancer cell lines.”.

Line 65: Consider “…isolated from samples of the intertidal seaweed…”

Line 66: Revise “Collected” to “collected”.

Line 84: As along the whole manuscript, the authors should use British English. Correct “neighbor” to “neighbour”.  

Concerning the Materials and Methods section, the authors should uniformize the identification of the suppliers as sometimes solely the supplier is mentioned, while in other cases the location is also detailed.

Line 190-191 (Waters Corp, Milford, MA, USA) / Line 245 (SIGMA).

The structure elucidation of desertomycin G is sufficiently detailed and the 1D and 2D NMR data seems to corroborate the proposed structure. Nevertheless, minor corrections should be performed since the NMR data indicates the presence of 19 oxygenated methines (and not 18) and 10 aliphatic methylenes (and not 8). Furthermore, captions from Figure S3 and S4 should be corrected since CD3OD has been used and not CDCl3.

Finally, the authors are advised to clarify why did they highlight the growth inhibitory effects towards Mycobacterium tuberculosis strains, since the most pronounced effects upon exposure to desertomycin G were observed towards another Gram-positive strains, particularly Corynebacterium urealyticum. Furthermore, it would be advisable (but not mandatory), to compare the obtained results with conventional antibacterial agents in order to unequivocally classify the observed effects as “strong” or “moderate”.

In conclusion, the results obtained with the current study are relevant and I recommend the acceptance of the manuscript, subject to tidying up the abovementioned issues.         

Author Response

With the current study (marinedrugs-437154), the authors report the isolation and structure elucidation of a new desertomycin as well as its antibacterial and cytotoxic effects towards a series of cancer cell lines. The content is unequivocally aligned with the scope of the journal, i.e. the structural novelty of the compound and the preliminary results on its potential therapeutic utility. The acceptance of the manuscript is advised, but minor changes and suggestions should be considered by the authors.

We appreciate these positive comments.

While generally well-written, very minor modifications should be considered as in:

Line 28: Consider “…isolated from samples of the intertidal seaweed Ulva sp.”

 Changed

Line 34: Enterococcus pyogenes should be deleted.

Deleted

Line 59-61: According to the authors it appears that bioactives-producing Streptomyces spp. are solely and restrictively obtained from marine seaweeds, which is scientifically incoherent, as they are frequently isolated from other sources such as terrestrial ones. As such, I would recommend modifying to: ….bioactive Streptomyces species are frequently associated to marine macro-algae”. 

We acknowledge this comment but have used sometimes instead of frequently. We do not want to give the idea that marine macro-algae are one of the major sources from where actinomycetes have been isolated because these microorganisms have also been found in many other marine sources.

Lines 62-63: The term “cytotoxic” solely reflects cell toxicity and not directly any potential therapeutic utility if not towards cancer cells. As such, I would recommend modifying to: “…of new natural products with antibiotic properties and cytotoxic activities towards cancer cell lines.”.

 Changed

Line 65: Consider “…isolated from samples of the intertidal seaweed…”

 Changed

Line 66: Revise “Collected” to “collected”.

 Changed

Line 84: As along the whole manuscript, the authors should use British English. Correct “neighbor” to “neighbour”.  

 Changed

Concerning the Materials and Methods section, the authors should uniformize the identification of the suppliers as sometimes solely the supplier is mentioned, while in other cases the location is also detailed.

Line 190-191 (Waters Corp, Milford, MA, USA) / Line 245 (SIGMA).

Missing information of suppliers has been added

The structure elucidation of desertomycin G is sufficiently detailed and the 1D and 2D NMR data seems to corroborate the proposed structure. Nevertheless, minor corrections should be performed since the NMR data indicates the presence of 19 oxygenated methines (and not 18) and 10 aliphatic methylenes (and not 8). Furthermore, captions from Figure S3 and S4 should be corrected since CD3OD has been used and not CDCl3.

We apologize for these mistakes.  We have corrected the number of oxygenated methines. 8 is the number of aliphatic methines, not included in the previous version, and we have decided to split the number of methylenes as 9 aliphatic and one nitrogenated. Captions of figures S3 and S4 have also been modified.

Finally, the authors are advised to clarify why did they highlight the growth inhibitory effects towards Mycobacterium tuberculosis strains, since the most pronounced effects upon exposure to desertomycin G were observed towards another Gram-positive strains, particularly Corynebacterium urealyticum. Furthermore, it would be advisable (but not mandatory), to compare the obtained results with conventional antibacterial agents in order to unequivocally classify the observed effects as “strong” or “moderate”.

The activity against M. tuberculosis has been highlighted due to the identification of this pathogen, especially its emerging resistant strains, as a major health threat by the WHO (500.000 cases of new infections/year). Although our compound is much more potent against C. urealyticum, the incidence of infections caused by this pathogen is much lower and they are not considered a public health problem. Nonetheless, the importance of the antibiotic potential of our compound against other strains is also highlighted in the discussion

Regarding the comment about the comparison between our results and those obtained with conventional antibacterial agents, please note that the profile of each strain is indicated in table 2. Most strains used are resistant versus one or more antibiotics, and these have been used as multiple controls to assess the performance of each pathogenic strain. In summary, the number of controls used is multiple and they are too numerous as to be included in the table.

Finally, the definition of the antibacterial activity as moderate or strong is rather ambiguous and there is no clear division between both classifications. We have somehow established in the first paragraph of section 2.3 the limit of 16 µg/mL to separate both categories, but for sure an unequivocal classification cannot be established for so different strains. In our hands, such MIC of 16 µg/mL can be very promising or not depending on the particular strain treated with the drug and its resistance profile.

In conclusion, the results obtained with the current study are relevant and I recommend the acceptance of the manuscript, subject to tidying up the abovementioned issues.

We acknowledge again the positive comments of this reviewer ad his/her suggestions to clearly improve the quality of the final version of our manuscript.